# Sofosbuvir Selects for Drug-Resistant Amino Acid Variants in the Zika Virus RNA-Dependent RNA-Polymerase Complex In Vitro

**DOI:** 10.3390/ijms22052670

**Published:** 2021-03-06

**Authors:** Adele Boccuto, Filippo Dragoni, Francesca Picarazzi, Alessia Lai, Carla Della Ventura, Carla Veo, Federica Giammarino, Francesco Saladini, Gianguglielmo Zehender, Maurizio Zazzi, Mattia Mori, Ilaria Vicenti

**Affiliations:** 1Department of Medical Biotechnologies, University of Siena, 53100 Siena, Italy; adele.boccuto@gmail.com (A.B.); dragonifilippo@gmail.com (F.D.); federica.giammari@gmail.com (F.G.); saladini6@unisi.it (F.S.); maurizio.zazzi@unisi.it (M.Z.); 2Department of Biotechnology, Chemistry and Pharmacy, University of Siena, 53100 Siena, Italy; francesca.picarazzi@student.unisi.it (F.P.); mattia.mori@unisi.it (M.M.); 3Department of Biomedical and Clinical Sciences L. Sacco, University of Milan, 20157 Milan, Italy; alessia.lai@unimi.it (A.L.); carla.dellaventura@gmail.com (C.D.V.); gianguglielmo.zehender@unimi.it (G.Z.); 4Clinical Pathology Analysis Laboratory ASL Taranto, 74023 Taranto, Italy; carlaveo@libero.it

**Keywords:** sofosbuvir, Zika virus, genetic barrier, in vitro selection, molecular dynamics

## Abstract

The nucleotide analog sofosbuvir, licensed for the treatment of hepatitis C, recently revealed activity against the Zika virus (ZIKV) in vitro and in animal models. However, the ZIKV genetic barrier to sofosbuvir has not yet been characterized. In this study, in vitro selection experiments were performed in infected human hepatoma cell lines. Increasing drug pressure significantly delayed viral breakthrough (*p* = 0.029). A double mutant in the NS5 gene (V360L/V607I) emerged in 3 independent experiments at 40–80 µM sofosbuvir resulting in a 3.9 ± 0.9-fold half- maximal inhibitory concentration (IC_50_) shift with respect to the wild type (WT) virus. A triple mutant (C269Y/V360L/V607I), detected in one experiment at 80 µM, conferred a 6.8-fold IC_50_ shift with respect to the WT. Molecular dynamics simulations confirmed that the double mutant V360L/V607I impacts the binding mode of sofosbuvir, supporting its role in sofosbuvir resistance. Due to the distance from the catalytic site and to the lack of reliable structural data, the contribution of C269Y was not investigated in silico. By a combination of sequence analysis, phenotypic susceptibility testing, and molecular modeling, we characterized a double ZIKV NS5 mutant with decreased sofosbuvir susceptibility. These data add important information to the profile of sofosbuvir as a possible lead for anti-ZIKV drug development.

## 1. Introduction

Zika virus (ZIKV) is an emerging human pathogen belonging to the *Flaviviridae* family, a group of arthropod-borne positive-sense single-stranded RNA viruses [1]. While ZIKV has been long known to be transmitted by the bite of *Aedes* spp. mosquitoes, additional transmission routes have been demonstrated in the last few years, including sex, blood transfusion, and vertical transmission [2,3,4,5]. After the first large outbreak in the Island of Yap in 2007, ZIKV spread to French Polynesia in 2013 and then to the Pacific Islands, eventually causing the last serious outbreak in Brazil and the Americas [6,7]. To date, a total of 86 countries have reported cases of mosquito-transmitted ZIKV infection and consequently, in 2016, the World Health Organization (WHO) declared ZIKV infection an international public health emergency [8].

Symptomatic ZIKV infection consists of nonspecific, flulike symptoms, such as cutaneous rash, arthralgia, and conjunctivitis [9]. In addition, during the recent epidemic, ZIKV infection has been associated with severe diseases, including multiorgan failure [10]; neurological complications, such as Guillain-Barré syndrome (GBS) in adults; and congenital ZIKV syndrome in newborns [11,12,13], possibly associated with increased virulence and neurotropism of the Asiatic lineage. The size of the epidemic and severity of the disease have renewed interest in the ZIKV infection, which can no longer be considered a benign disease [9,14]. 

Despite the clinical relevance of the ZIKV infection, at present, there are neither ZIKV-specific antivirals drugs nor vaccines [15]. Currently, there are different clinical trials testing at least 16 ZIKV vaccine candidates, (www.who.int/immunization/research/vaccine_pipeline_tracker_spreadsheet/en/ (accessed on 5 March 2021)), however, vaccine development is challenged by safety concerns, due to the risk of vaccine-associated GBS and the enhancement of diseases with other endemic flaviviruses [16]. Candidate targets for anti-ZIKV drugs include viral proteins such as protease located in the NS3 viral gene and RNA-dependent RNA polymerase (RdRp) located in the NS5 viral gene as well as host targets used during viral entry and replication [17,18]. To address the urgent need for anti-ZIKV therapy, repurposing of licensed antivirals is under evaluation [19,20,21,22]. Given the high degree of NS5 homology observed among members of the *Flaviviridae* family [23,24], sofosbuvir, a licensed uridine nucleotide analogue widely used for highly effective and safe treatment of the hepatitis C virus (HCV) infection [25], has been recently evaluated as an anti-Flavivirus lead candidate. The inhibitory activity of sofosbuvir against different flaviviruses as well as against the Chikungunya virus has been well documented in vitro and in vivo [26,27,28,29,30,31,32]. Noteworthily, sofosbuvir showed a protective effect of neuronal stem cells (NCs) from ZIKV and inhibition of vertical ZIKV transmission in mouse models [33,34] and in rhesus monkeys [35]. In addition, sofosbuvir has shown a high genetic barrier to resistance with HCV, both in vitro and in vivo, as a key component of its prolonged efficacy [36,37,38,39]. However, the in vitro selection of resistance mutations with Flaviviruses has been characterized only for West Nile virus (WNV) [27]. In this study, we investigated the ZIKV resistance profile against sofosbuvir through cell-based in vitro selection experiments. 

## 2. Results

### 2.1. ZIKV In Vitro Selection Experiments under Sofosbuvir Drug Pressure 

As described in Section 4.4, two ZIKV viral inputs at multiplicity of infection (MOI) 0.01 and 0.05, each in duplicate, were used to infect Huh7 cells in the presence of increasing sofosbuvir concentrations, starting from 5 µM, corresponding to 2-fold (2.5 ± 0.6 µM) sofosbuvir half-maximal inhibitory concentration (IC_50_), with the wild type (WT) virus. Uninfected cells plus sofosbuvir were used as a reference to discriminate the virus-induced cytopathic effect (CPE) from sofosbuvir cytotoxicity and physiological cell mortality. Drug pressure significantly delayed viral growth with respect to the no-drug control virus (CV), and the time for viral breakthrough increased with increasing drug concentrations (*p* = 0.0286). Indeed, 9-, 15- and 22-days post infection (dpi) were required to achieve 80% CPE at 5, 10, and 20 µM sofosbuvir, respectively. At 40 µM, samples were harvested after 40.7 ± 9.4 dpi, whereas at the highest concentration tested (80 µM), samples needed 20.5 ± 2.4 dpi for viral rebound. By contrast, the virus control was consistently collected at 4.2 ± 1.9 dpi. The two different MOIs used did not affect the time of viral rebound and no toxicity was observed at the sofosbuvir concentrations tested. *In vitro* resistance selection experiments were stopped after a mean of 107.3 ± 8.5 dpi when the drug pressure was 32-fold higher than starting sofosbuvir IC_50_.

No aminoacidic variations with respect to the WT NS5 sequence were observed with 5, 10, and 20 µM sofosbuvir by either Sanger sequencing or Next Generation Sequencing (NGS). At 40 µM sofosbuvir, the double NS5 mutant V360L/V607I was selected in experiments 2 and 5 and the H289Y substitution was selected in experiment 4 only by NGS. At 80 µM sofosbuvir, the double mutant V360L/V607I was maintained in experiments 2 and 5 and also emerged in experiment 1. In addition, the strain in experiment 5 acquired the C269Y mutation, while the H289Y substitution was maintained in experiment 4, as determined by both NGS and Sanger sequencing. Thus, the V360L and V607I mutations were selected in three separate experiments independently by the different initial virus input used. The emergence of the double mutant was associated with a reduction in the time for viral breakthrough from 40.7 ± 9.4 to 20.5 ± 2.4 dpi. None of the NS5 mutations selected in vitro were retrieved in a dataset of 562 sequences obtained from circulating African and Asian ZIKV lineages (https://www.ncbi.nlm.nih.gov/genomes/VirusVariation (accessed on 20 January 2021)) [40], excluding their possible pre-existence as virus natural polymorphisms. 

To assess whether emergent NS5 mutations were associated with reduced drug susceptibility, sofosbuvir IC_50_ was measured against the viral stocks collected at 40 µM and 80 µM as well as the corresponding WT viruses, to exclude fold change (FC) variation independent of the NS5 substitutions. As indicated in Table 1, no changes in FC were observed in the absence of the NS5 mutations. In the presence of the double mutant (V360L/V607I), IC_50_ values consistently increased with respect to the paired WT CV (median FC 4.0 [3.3–5.1 IQR]). The maximum increase in FC (6.8) was observed in experiment 5, at 80 µM sofosbuvir, where the double mutant was associated with the C269Y substitution. The H289Y mutation did not appear to be involved in drug resistance on its own, causing a minimal shift in FC at 40 µM (1.1) and 80 µM (1.9).

Interestingly, amino acid substitutions were also detected in the NS3 and NS4B proteins, which form the replication complex together with NS5 polymerase (Table 1 and Table 2). The NS3 T27S variant emerged in association with the NS5 double mutant and the NS4B substitutions Q172Y, V173L, L175S and R187Q (experiment 2 with 40 µM sofosbuvir, FC 4.0). Increasing drug pressure to 80 µM, the virus selected in experiment 2 lost the NS4B substitutions and maintained a similar FC (5.1) but reduced the time to viral breakthrough with respect to the same experiment at 40 µM (Table 1), suggesting a role for NS4B variations in viral fitness rather than in sofosbuvir resistance. Similarly, the NS4B Y87H variant emerging in experiment 4 in association with H289Y did not result in any FC shift. The NS4B Q172R mutation emerged in experiment 5 together with the C269Y, V360L, and V607I substitutions in NS5 (FC 6.8). 

### 2.2. Molecular Modeling

Due to the lack of structural information on the ZIKV RdRp/sofosbuvir interaction, the possible binding mode of sofosbuvir triphosphate [41] and its nucleotide analogue, UTP, to the catalytic site of the WT and V360L/V607I mutant ZIKV RdRp was investigated by molecular docking and molecular dynamics (MD) simulations. A visual inspection of representative structures extracted from MD trajectories showed that, in both WT and mutant ZIKV RdRp, the uracil ring from UTP (Figure 1A,B) and sofosbuvir (Figure 1C,D) is base-paired to A from the RNA template in a Watson–Crick conformation, while the phosphate groups are coordinated to Mg^2+^ ions, lysine, and arginine residues. Theoretical affinity of sofosbuvir and UTP for ZIKV RdRp was computed as the delta energy of binding (ΔE_b_) by the Molecular Mechanics Generalized Born Surface Area (MM-GBSA) approach [42]. Results showed that the V360L/V607I mutant has a limited or negligible effect on the theoretical affinity of sofosbuvir and UTP to the RNA template, when they are paired to A. To understand the molecular bases of the decreased efficacy of sofosbuvir observed experimentally in the V360L/V607I mutant, we hypothesized that the drug might be included in the nascent RNA regardless of the nucleotide in the RNA template, based on the relatively low fidelity of RdRp in nucleotide insertion [43]. To this aim, the binding mode of sofosbuvir was also investigated when paired to C, G, and U in the RNA template, using the same computational procedure described above. A significant decrease of sofosbuvir theoretical affinity was observed (Table 3) when the pairing base in the RNA template of the V360L/V607I mutant RdRp was G or U. Systems in which C is opposed to sofosbuvir represent an exception, because sofosbuvir showed a slightly stronger affinity for the mutant compared to WT RdRp. This is probably due to the inclusion of a water molecule in the catalytic site of mutant RdRp, which mediates the interaction between sofosbuvir and C from the RNA template, and the H-bond interaction between the uracil ring of sofosbuvir and Lys458 (Figure 2B). In contrast, in the WT RdRp, the H-bond to Lys458 is lost (Figure 2A). However, compared to other systems, the affinity of sofosbuvir when paired to C in the RNA template is significantly lower. In WT systems with G and U in the RNA template (Figure 2C,E), sofosbuvir pairs to the opposite nucleotide in a wobblelike conformation while also H-bonding to Lys458, whereas in the corresponding mutant forms (Figure 2D,F), an H-bond interaction from this network is lost, which might explain the decrease of sofosbuvir theoretical affinity.

## 3. Discussion

In the absence of specific antiviral treatment, ZIKV therapy remains only supportive and not adequate to mitigate potentially severe disease. The systematic screening of FDA-approved antivirals has demonstrated anti-ZIKV activity for compounds targeting well-conserved functions across different viruses [19,20,44,45]. Recent data have suggested a possible use of sofosbuvir, a potent and safe HCV polymerase inhibitor, for the treatment of ZIKV infection [30]. While the antiviral activity of sofosbuvir against ZIKV has been well characterized, its resistance profile has not yet been elucidated. Only one study [46] showed decreased sofosbuvir activity in a cell-free biochemical assay against a ZIKV polymerase construct carrying the S604T substitution corresponding to the well-characterized S282T mutation located in the HCV NS5B sequence and conferring resistance to sofosbuvir as an anti-HCV agent. While this study supports the analogy between ZIKV and HCV RdRp in their interaction with sofosbuvir, it does not provide any clue to sofosbuvir resistance selection in ZIKV RdRp, a key component of drug profiling. In the present study, Huh7 cells were infected in duplicate with two different inputs of viral stock (MOI 0.01 and 0.05) and exposed to increasing sofosbuvir concentrations, starting with 5 µM, 2-fold above the sofosbuvir IC_50_. The in vitro resistance selection experiments were stopped when all the cultures showed 80% CPE at the maximum sofosbuvir concentration tested (80 µM, corresponding to 32-fold higher than the IC_50_ in the same system), occurring at a mean ± SD of 107.3 ± 8.5 dpi. 

The increased time required for viral breakthrough at increasing sofosbuvir concentration indicates that the drug actively inhibits ZIKV replication. Although in vitro systems are not directly comparable, the time to emergence of sofosbuvir-resistant mutants appears to be similar with ZIKV (95 ± 20 days in this study) and HCV (90 ± 14 days in the work showing selection of the S282T substitution) [47]. By contrast, using the same approach adopted in the current study, we previously reported a significantly shorter time to selection of sofosbuvir resistance with WNV (49 ± 0 days). The genetic barrier to resistance to sofosbuvir during HCV treatment in vivo is known to be high, however, the similarity between HCV and ZIKV in vitro data is not sufficient to assume that the same occurs in vivo. It must also be noted that in vitro selection experiments do not mimic in vivo viral sequence variability, therefore, additional mutational patterns selected in vivo may emerge and contribute to resistance.

To escape sofosbuvir pressure, the ZIKV NS5 region consistently acquired the amino acid substitutions V607I and V360L, in experiments 2 and 5 at 40 µM sofosbuvir and in experiment 1 at 80 µM. This mutational pattern conferred reduced susceptibility to sofosbuvir, as measured by a mean 3.9 ± 0.9-fold IC_50_ shift with respect to the WT virus. The highest IC_50_ FC (6.8) was detected with the triple mutant C269Y/V607I/V360L. Interestingly, the V607I mutation, corresponding to position 286 on HCV NS5B, is located at the junction of the fingers and palm domains in motif B, a well-conserved domain among RNA polymerases from positive-sense RNA viruses [48]. The RdRp motif B determines the nucleotide choice and is implicated in resistance to nucleotide inhibitors. In HCV, amino acid residues K51, S282, T286, and M289 are the closest to the active site of polymerase, with S282 being the key residue in contact with the substrate [47]. With HCV, S282T frequently emerged in in vitro sofosbuvir resistance selection experiments, decreasing susceptibility from 2.4 to 19.4-fold [39,49], and emerged in vivo in the Phase II ELECTRON clinical trial [37]. In addition, the WNV NS5 S604T mutation, corresponding to HCV NS5B S282T, was acquired by WNV in vitro during resistance selection experiments at 80 µM sofosbuvir, further corroborating the role of the SGxxxT domain for sofosbuvir activity [27]. Considering the high degree of conservation of the SGxxxT consensus domain among flaviviruses, a role in resistance to sofosbuvir is possible for substitutions close to the key serine as in the case of V607I detected in our in vitro selection experiments. The V360L mutation, selected in combination with V607I, occurred within the RdRp nuclear localization sequence (NLS), which comprises the βNLS domain (residues 316–367) lying on top of the thumb subdomain and the α/βNLS domain (residues 368–415) located between the fingers and the palm subdomain [50]. In our in vitro selection experiments, V360L consistently appeared in combination with the V607I substitution in multiple experiments, making it impossible to distinguish a direct impact on the increase of sofosbuvir IC_50_ from a compensatory role to restore enzyme function. In MD simulations, the double mutant V360L/V607I impacts the binding mode of sofosbuvir, suggesting a role in sofosbuvir resistance. Interestingly, when sofosbuvir was paired to A from the RNA template, the V360L/V607I mutation did not affect the binding mode of the drug nor its theoretical affinity. In contrast, when it was paired to G and U, the V360L/V607I mutation perturbed the interaction network of sofosbuvir with a corresponding decrease of its theoretical affinity compared to WT RdRp. Pairing with C provides the lower theoretical affinity for WT RdRp with a value that is comparable to sofosbuvir/A pairing in the V360L/V607I mutant. Considering that sofosbuvir is a uridine analogue, this supports a relevant role for sofosbuvir binding to nucleotides different from A during RNA extension. 

Similar to V360L, C269Y did not appear alone in in vitro selection experiments and it was not possible evaluate its role in sofosbuvir resistance. C269 is located in the poorly conserved 10-residue linker domain (amino acids 263 to 272), which is essential for the adequate interaction between the MTase and RdRp domains, resulting in the production of infectious viral particles [50,51]. However, the triple mutant C269Y/V360L/V607I was not investigated in silico due to the lack of reliable structural data and because C269 is located in a highly flexible region far from the catalytic site. The H289Y mutation emerged only in experiment 4 at 40 µM and at 80 µM sofosbuvir and did not appear to impact sofosbuvir activity (FC 1.1 and 1.9, respectively). Position 289 is located in the first part of RdRp domain and has been suggested to be involved in RdRp stabilization [52]. Moreover, H289 is replaced by a glutamine (Genbank accession number KU922960; KU922923) or a lysine (Genbank accession number KU44693) in some clinical isolates. The role of these mutations remains to be elucidated [53]. Thus, the emergence of H289Y, similar to the emergence of other amino acid substitutions in regions other than NS5 (Table 1 and Table 2), could be related to improved viral fitness rather than resistance. It must be noted that some mutations were transiently or permanently observed in the absence of drug pressure, suggesting that changes in the viral genome driven by virus adaptation to the cell line used or stochastically do also occur (Table 2).

In conclusion, this work characterized sofosbuvir activity against ZIKV in vitro and described the selection of resistance for the first time. Although sofosbuvir IC_50_ is higher against ZIKV compared to HCV (ranging from 0.1 to 5 µM for ZIKV and from 0.01 to 0.1 µM for HCV), sofosbuvir antiviral activity against ZIKV has been demonstrated in immortalized and primary human cell lines as well as in murine models where it reduced viremia and decreased the rate of transmission from mother to fetus [34]. Selection of sofosbuvir resistance in vitro corroborates antiviral activity against ZIKV and may not be a major problem in short-term treatment in vivo. Despite the low micromolar activity, sofosbuvir can be envisioned as a lead in ZIKV therapy, benefitting from clinical safety data accumulated from HCV therapy—particularly in pregnant women [54]—and providing a chance for treatment of severe cases of ZIKV infections [55], possibly reducing the occurrence of birth defects. 

## 4. Materials and Methods

### 4.1. Cells and Virus

The African green monkey kidney cell line Vero E6 (ATCC catalog no. CRL-1586) and the human hepatoma cell line Huh7 (kindly provided by Istituto Toscano Tumori, Core Research Laboratory, Siena, Italy) were cultured in high-glucose Dulbecco’s Modified Eagle’s Medium with sodium pyruvate and L-glutamine (DMEM; Euroclone, Milan, Italy), supplemented with 10% Fetal Bovine Serum (FBS; Euroclone, Milan, Italy) and 1% Penicillin/Streptomycin (Pen/Strep, Euroclone, Milan, Italy), and incubated at 37 °C in a humidified incubator supplemented with 5% CO_2_. The same medium was used but with a lower FBS concentration for viral propagation and drug susceptibility testing (1%) and for in vitro selection experiments (3%).

The H/PF/2013 ZIKV strain belonging to the Asian lineage (GenBank Sequence Accession Number KJ776791) was kindly provided by the Istituto Superiore di Sanità, Rome, Italy. Once expanded, ZIKV viral stock was titrated in VERO E6 cells by plaque assay (PA) as described by Vicenti et al. [31], yielding 4.2 × 10^5^ PFU/mL. 

### 4.2. Drug and Cytotoxicity Assay

The FDA-approved anti-HCV compound sofosbuvir (β-d-2′-deoxy-2′-α-fluoro-2′-β-C-methyluridine; MCE^®^ cat. HY-15005) was supplied as powder and dissolved in 100% dimethyl sulfoxide (DMSO). Before each assay, sofosbuvir was properly diluted to reach the desired working concentration. The sofosbuvir 50% cytotoxic concentration (CC_50_) was measured in Huh7 cells by CellTiter-Glo^®^ 2.0 luminescent cell viability assay (Promega, Madison, WI, USA) according to the manufacturer’s protocols. After 72 h incubation, the luminescence values obtained from cells treated with sofosbuvir or DMSO were measured through the GloMax^®^ Discover Multimode Microplate Reader (Promega Madison, WI, USA) and elaborated with the GraphPad PRISM software version 6.01 (La Jolla, CA, USA) to calculate the CC_50_. 

### 4.3. Determination of Sofosbuvir Antiviral Activity by Immunodetection Assay (IA)

Titration of the viral stock and determination of sofosbuvir antiviral activity were performed in the Huh7 cell line by immunodetection of viral antigen as previously described [56]. Briefly, preseeded Huh7 cells in 96-well plates were adsorbed for 1 h at 37 °C with 50 TCID_50_ of ZIKV viral stock. After removal of virus inoculum, serial dilutions of sofosbuvir were added to the cells and the plates were incubated for 72 h at 37 °C with 5% CO_2_. For the immunodetection assay (IA), cells were fixed for 30 min with 10% formaldehyde (Carlo Erba, Milan, Italy), rinsed with 1% PBS, and permeabilized for 10 min with 1% Triton X-100 (Carlo Erba Milan, Italy). After washing with PBS containing 0.05% Tween 20 (Carlo Erba, Milan, Italy), plates were incubated for 1 h with monoclonal anti-flavivirus mouse antibody (clone D1-4G2-4-15; Novus Biologicals, Centennial, CO, USA) diluted 1:400 in blocking buffer (PBS containing 1% BSA and 0.1% Tween 20). After washing, cells were incubated for 1 h with a polyclonal Horseradish Peroxidase (HRP)-coupled antimouse IgG secondary antibody (NB7570, Novus Biologicals, Centennial, CO, USA) diluted 1:10,000 in blocking buffer. Next, cells were washed and the 3,3′,5,5′-Tetramethylbenzidine substrate (TMB, Sigma Aldrich, St. Louis, MO, USA) was added to each well. After the addition of 0.5 M sulfuric acid, absorbance was measured at 450 nm optical density (OD450) using the Absorbance Module of the GloMax^®^ Discover Multimode Microplate Reader (Promega, Madison, WI, USA) and adjusted by subtracting the background value established as twofold the mean OD_450_ value of quadruplicate uninfected cells. Each IA run was validated by the OD_450_ value above 1 in the virus control culture. OD_450_ values from each well were normalized according to the 100% and 0% of viral replication and normalized values were used to calculate half-maximal inhibitory concentration (IC_50_) values through a nonlinear regression analysis of the dose-response curves generated with GraphPad PRISM software version 6.01. 

### 4.4. In Vitro Selection Experiments

*In vitro* selection experiments were performed as previously described [27]. Briefly, Huh-7 cells at 70% confluence in T25 flasks were infected with virus at 0.05 and 0.01 MOI, each in duplicate. After 1 h adsorption at 37 °C with 5% CO_2_, supernatants were removed and cells were treated with an initial concentration of 5 µM sofosbuvir, corresponding to 2-fold the IC_50_. Cells were incubated and monitored every 24 h and when the CPE affected approximately 80% of the cells, the supernatants were harvested. In order to obtain a higher viral titer, cells were subjected to one cycle of freezing and thawing, then, cellular debris were cleared through centrifugation for 30 min at 1300 g and viral stocks were stored at −80 °C. Subsequent passages were set up using 2 mL of the harvested virus to infect a new culture of Huh7 cells in the presence of a 2-fold higher concentration of sofosbuvir (10, 20, 40, and 80 µM). Infected cells without sofosbuvir (CV) and uninfected cells supplemented with sofosbuvir (control cells, CC) were included at each passage. Sanger sequencing (Section 4.5) of the ZIKV NS5 region and next-generation sequencing (Section 4.6) of the most relevant ZIKV genes (NS1, NS2A, NS2B, NS3, NS4A, NS4B, and NS5) were performed to detect emergent mutations at each virus breakthrough with increasing sofosbuvir concentration. Mutant viruses were titrated by IA, and sofosbuvir IC_50_ was measured as described in Section 4.3.

### 4.5. Viral RNA Amplification and Sequencing

Total RNA was extracted in duplicate from 150 μL of viral stocks derived from in vitro selection experiments, using the ZR Viral RNA Kit (Zymo Research, Irvine, CA, USA) according to the manufacturer’s protocol. The complementary DNA (cDNA) was generated by random hexamer-driven reverse transcription using 10 µL of heat-denaturated RNA extract, 660 µM dNTPs, 6 μL 5X ImProm-II^TM^ Reaction Buffer, 50 ng hexanucleotides, 1.5 mM MgCl_2_, 20 U RNasin^®^ Plus RNase Inhibitor, and 1 U of ImProm-II™ Reverse Transcriptase (Promega, Madison, WI, USA), for a final volume of 30 μL. Reactions were run in an Eppendorf Mastercycler Nexus (Eppendorf, Hamburg, Germany) apparatus for 30 min at 37 °C followed by enzyme inactivation for 5 min at 80 °C. The cDNA was used as the template to amplify the whole NS5 gene in multiple PCR reactions. To design primers with a high degree of conservation, the ZIKV alignment available at the National Centre for Biotechnology Information (NCBI) site and representative of all the circulating ZIKV strains was used (https://www.ncbi.nlm.nih.gov/genomes/VirusVariation (accessed on 20 January 2020)) [40]; primer sequences with coordinates referred to the H/PF/2013 ZIKV strain (GenBank KJ776791) are indicated in Table 4. Primers were synthesized by the Eurofins Genomic (Ebersberg, Germany). The PCR mixture included 3 μL cDNA, 10 µL 5× Q5 Reaction buffer (NEB, Ipswich, MA, USA), 10 pmol P823 and P824, 320 µM dNTPs, and 1 U Q5 Hot Start High-Fidelity DNA Polymerase (NEB), for a final volume of 50 μL. Reactions were performed in an Eppendorf Mastercycler Nexus (Eppendorf, Hamburg, Germany) with an initial denaturation step at 98 °C for 3 min followed by 40 cycles each including 30 s at 68 °C, 2 min at 72 °C, and 10 s at 98 °C and a final step at 72 °C for 5 min.

Bidirectional DNA sequencing reactions were performed using the BrilliantDye^TM^ Terminator Kit v1.1 (NimaGen, Nijmegen, 

The Netherlands) with nine different primers spanning the whole NS5 region (Table 4). Briefly, 3 uL of PCR products, diluted at a final concentration of 1–3 ng/μL, were mixed with 3.2 pmol/μL of each sequencing primer, 0.5 μL of BrilliantDye^TM^ Terminator Ready Reaction Sequencing, and 2 μL of 5× Sequencing Buffer, for a final volume of 10 uL. The reactions were denatured at 96 °C for 1 min followed by 25 cycles at 50 °C for 5 s, 60 °C for 4 min, and 96 °C for 10 s. Sequencing reactions were treated with X-Terminator^®^ Purification kit (Applied Biosystems, Waltham, MA, USA) in a 96-well plate as suggested by the manufacturer, then resolved by capillary electrophoresis with the 3130 XL Genetic Analyzer (Applied Biosystems, Waltham, MA, USA). Chromatograms were assembled and edited with the DNAStar 7.1.0 SeqMan module. All NS5 sequences generated from viral populations emerged during in vitro selection experiments were aligned with the WT virus strain and with the reference sequences obtained from Virus Variation NCBI database (https://www.ncbi.nlm.nih.gov/genomes/VirusVariation/Database/nph-select.cgi (accessed on 1 November 2019)) [40], in order to identify variation in NS5 sequences driven by drug pressure.

### 4.6. Next Generation Sequencing

ZIKV RNA was reverse-transcribed using the random primer FR26RV-N (10 µM). The first-strand cDNA obtained was denatured at 94 °C for 3 min, then chilled on ice for 2 min, and 5 U of Klenow fragment (New England Biolabs, Ipswich, MA, USA) was directly added to the reaction to perform second-strand cDNA synthesis for 1 h at 37 °C and 10 min at 75 °C. Next, 5 µL of double stranded DNA (dsDNA) was added to the PCR master mix containing 4 µL of 10× AccuPrime PCR buffer I, 0.2 µL of AccuPrime Taq DNA Polymerase high fidelity, 4 µL of 10 µM FR20RV, and 35.8 µL of water; the reaction was incubated at 94 °C for 2 min, then at 94 °C for 30 s and 55 °C for 1 min for 40 cycles, and finally at 68 °C for 3 min [57]. To enrich the coverage of NS5 sequencing, RNA was subjected to cDNA synthesis using the SuperScript^TM^ III First-Strand Synthesis SuperMix for qRT-PCR (Invitrogen Life Technologies, Carlsbad, CA, USA) and amplified by means of 5 primer pairs amplifying 5 overlapping genome fragments [58] with GoTaq Hot Start Polymerase (Promega, Madison, WI, USA). Five microliters of dsDNA were added to the PCR master mix containing 10 µL of 5× Green GoTaq^®^ Flexi Buffer, 4 µL dNTPs, 3 µL MgCl_2_, 0.5 µL of 50 µM of each primer, and 26.5 µL of water, for a final volume of 50 µL. The thermal conditions were 94 °C for 2 min, 45 cycles of 94 °C for 1 min, 50 °C for 1 min, 72 °C for 1 min, and finally, 72 °C for 10 min. Ten microliters of PCR product were analyzed on 1.5% agarose gel stained with ethidium bromide. The amplicons were cleaned up using the QIAquick PCR Purification kit (QIAGEN, Hilden, Germany) in accordance with the manufacturer’s protocol and eluted into a final volume of 50 μL of distilled water. All the purified products were quantified using the Invitrogen Quant-iT Picogreen dsDNA assay.

Library preparation for Illumina sequencing was done using a Nextera^®^ XT DNA Sample Preparation and Index kit (Illumina, San Diego, CA, USA) according to the manufacturer’s manual. Resulting libraries were normalized and pooled for subsequent sequencing on an Illumina MiSeq platform using the 2 × 150 cycle paired-end sequencing protocol. FASTQ files were generated from MiSeq Reporter (Illumina, San Diego, CA, USA) and the paired reads were imported to Geneious software v10.1.3 (https://www.geneious.com/ (accessed on 1 June 2020)). Results were mapped and aligned to the reference Zika virus strain H/PF/2013 (accession number KJ776791.2). The mean coverage of full genomes, taking into account the results of random and specific primers, ranged from 50.5 to 3499.6. Minority species with a frequency above 5% were considered for the analysis.

### 4.7. Molecular Modeling

Due to the lack of a catalytically competent ZIKV RdRp experimental structure in the Protein Data Bank [59], the homology models of RNA-bound WT and V360L/V607I mutant of ZIKV RdRp were generated by homology modeling with the Prime software (Schrodinger Maestro, release 2019-4) [60]. In the C269Y/V360L/V607I triple mutant identified in vitro, Cys269 is in a highly flexible region far from the catalytic site. Available experimental structures describing Cys269 do not bind RNA and Mg^2+^ ions, reasonably representing a catalytically inactive conformation and being unsuitable templates for homology modeling. Therefore, this triple mutant was not investigated in silico. The ZIKV RdRp WT sequence was retrieved from the Uniprot database [61] (entry Q32ZE1) and used as a query, while the highest-sequence homology was observed in WNV RdRp [18]. The catalytically competent RNA-bound WNV RdRp model generated and validated in a previous work [27] was thus used as a structural template in the homology modeling of all the ZIKV RdRp structures described in this work, including WT and mutant RdRp as well as the variants in the RNA template sequence. Homology models were optimized through energy minimization in a rectilinear box of explicit three-site model (TIP3P) water molecules with the addition of counter-ions to neutralize the total charge. First, the solvent was relaxed for 500 steps using the Steepest Descent algorithm (SDA), followed by 1500 steps with the Conjugate Gradient algorithm (CGA). Then, each solvated system was energy minimized for 1500 steps with the SDA and 8500 steps with the CGA. The reliability of the homology models was assessed by comparison with one of the models previously generated by Šebera et al. [62] and kindly provided by the authors, giving an RMSD value of 1.9 Å for the catalytic site and 3.0 Å for the whole protein. Molecular docking simulations of sofosbuvir and UTP within the catalytic site of WT and V360L/V607I mutant ZIKV RdRp were carried out by the GOLD program using the CHEMPLP fitness function [63]. The protonation state of sofosbuvir and UTP was assigned with Openeye QUACPAC version 2.0.0.3 and energy minimized by Szybki version 1.10.0.3 (OpenEye Scientific Software; http://www.eyesopen.com (accessed on 1 October 2020)). Based on the sofosbuvir binding mode in the catalytic site of HCV RdRp, as described in the PDB ID 4WTG [64], the most reliable poses from docking simulations were relaxed by molecular dynamics (MD) simulations with the Amber18 program https://ambermd.org/ [65]. Docking complexes were solvated and energy minimized with the settings reported above. Then, temperature was gradually raised from 0 to 300 K for 1 ns using the Langevin thermostat (NVT ensemble), while 1 ns of density equilibration was carried out with the Berendsen barostat (NTP ensemble). A first equilibration of 50 ns was carried out, and finally, trajectories were produced for 500 ns (NPT ensemble). MD trajectories were analyzed with the CPPTRAJ software [66]. Theoretical affinity was computed by the Molecular Mechanics Generalized Born Surface Area (MM-GBSA) approach [42] over 100 frames of each trajectory using the single trajectory approach.

### 4.8. Statistical Analysis

Results of replicate antiviral activity measurements were reported as mean and standard deviation (SD). The difference in time for viral growth under different experimental conditions was analyzed by Mann–Whitney U-test. Statistical analysis was performed using GraphPad PRISM software version 6.01.

## Figures and Tables

**Figure 1 ijms-22-02670-f001:**
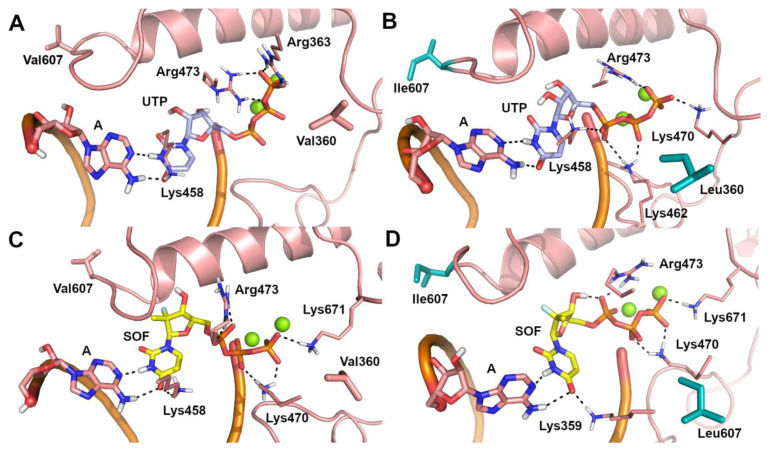
Predicted binding poses of UTP (light blue sticks) and sofosbuvir (SOF) (yellow sticks) within the catalytic site of ZIKV RNA dependent RNA polymerase (RdRp), in the representative structure extrapolated from molecular dynamics (MD) trajectories. (**A**) A-UTP-WT ZIKV RdRp system. (**B**) A-UTP-V360L/V607I mutant ZIKV RdRp system. (**C**) A-SOF-WT ZIKV RdRp system. (**D**) A-SOF-V360L/V607I mutant ZIKV RdRp system. Polar contacts are highlighted by black dashed lines. Residues involved in polar and non polar interactions with UTP and sofosbuvir are shown as sticks and are labelled. The opposite A from the RNA template is shown as sticks. Mg^2+^ ions are shown as green spheres. Mutated residues are highlighted by cyan sticks.

**Figure 2 ijms-22-02670-f002:**
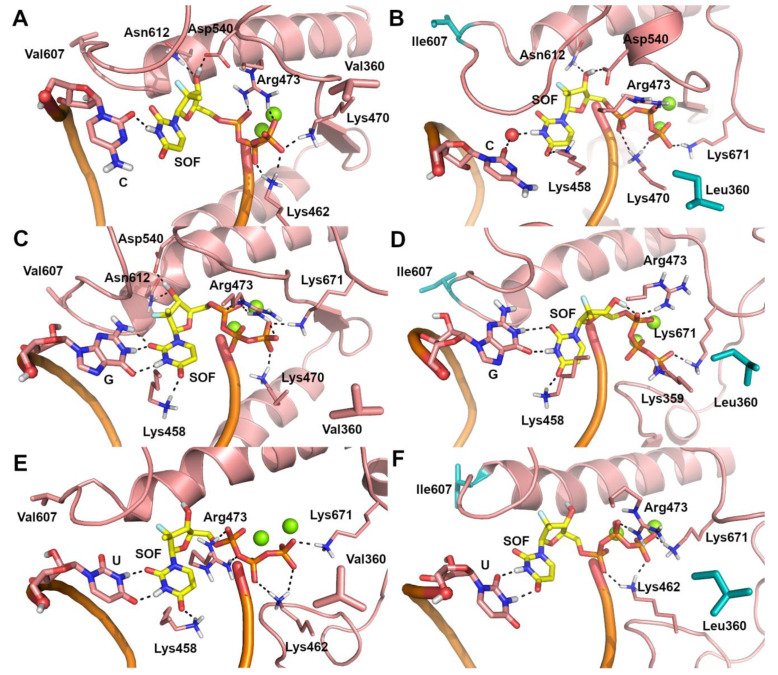
Predicted binding poses of sofosbuvir (SOF) (yellow sticks) within the catalytic site of ZIKV RdRp in the representative structures extrapolated from MD trajectories. (**A**) C-SOF-WT ZIKV RdRp system. (**B**) C-SOF-V360L/V607I mutant ZIKV RdRp system. (**C**) G-SOF-WT ZIKV RdRp system. (**D**) G-SOF-V360L/V607I mutant ZIKV RdRp system. (**E**) U-SOF-WT ZIKV RdRp system. (**F**) U-SOF-V360L/V607I mutant ZIKV RdRp system. Polar contacts are highlighted by black dashed lines. Residues involved in polar and non polar interactions with sofosbuvir are shown as sticks and are labelled. The opposite C, G, and U from the RNA template are shown as sticks. Mg^2+^ ions are shown as green spheres. Mutated residues are highlighted by cyan sticks.

**Table 1 ijms-22-02670-t001:** Reduction of sofosbuvir activity in mutated viral stocks at 40 and 80 µM drug pressure. In bold are indicated the mutations detected both by Sanger sequencing and by Next Generation Sequencing (NGS). Exp—experiment; MOI—Multiplicity of infection; IC_50_—half-maximal inhibitory concentration, each value is expressed in µM and is the mean of triplicate assays ± standard deviation; FC—Fold change; dpi—days post infection; VC—virus control. No variations with respect to the wild type (WT) sequence were detected in NS4A and NS2B regions, thus, it is not indicated in the table.

			Sofosbuvir 40 µM								Sofosbuvir 80 µM							
Experiment	Sofosbuvir Pressure	MOI	dpi	NS1	NS2A	NS3	NS4B	NS5	IC_50_	FC	DPI	NS1	NS2A	NS3	NS4B	NS5	IC_50_	FC
1	Yes	0.01	54						3.4 ± 1.3	0.7	18				Y87H	**V360LV607I**	15.0 ± 8.6	3.3
2	Yes	0.01	34	E315G		T27S	Q172Y V173L L175S R187Q	**V360L V607I**	18.6 ± 6.3	4.0	22	E315G		T27S		**V360L V607I**	23.0 ± 3.7	5.1
3^vc^	No	0.01	3						4.7 ± 0.7	1.0	3						4.5 ± 0.4	1.0
4	Yes	0.05	34		F9L		Y87H	H289Y	4.9 ± 0.7	1.1	19		F180L		Y87H	**H289Y**	5.0 ± 0.3	1.9
5	Yes	0.05	41		N98S		Y87H	**V360L V607I**	14.0 ± 5.9	3.0	23	K227C			Y87H Q172R	**C269Y, V360L V607I**	18.4 ± 5.3	6.8
6^vc^	No	0.05	3						4.6 ± 1.2	1.0	3			E371D			2.7 ± 1.1	1.0

**Table 2 ijms-22-02670-t002:** Whole Zika virus (ZIKV) aminoacidic substitutions emerged in vitro under sofosbuvir drug pressure in NS1, NS2A, NS3, NS4A, NS4B, and NS5 genes. **(c)**—Drug concentration; VC—virus control.

Experiment	Sofosbuvir (c)	NS1	NS2A	NS3	NS4A	NS4B	NS5
1	5 µM	K245T					
2	5 µM			V72I *			
3	VC matched with 5 µM	K245T	T176D C177I	V72I * L435R			
4	5 µM	K245T	T176D C177I	V72I * L435R			
5	5 µM	K245T	L17AP T176D C177I	V72I *			
6	VC matched with 5 µM	K245T	T176D, C177I/F	V72I * L435R			V832D * T833I K834C
1	10 µM	K245T					
2	10 µM						
3	VC matched with 10 µM	K245T	T176D C177I	V72I * L435R			
4	10 µM	K245T	V15A				
5	10 µM		T176N C177I		A103D		
6	VC matched with 10 µM	K245T	T176D C177I	L435R		T188L	
1	20 µM	K245T	L174S T176D C177I				V832D * T833I
2	20 µM	E315G	T176A C177F C177S				
3	VC matched with 20 µM	K245T		L189V L435R			
4	20 µM	K245T	C177I F180L				
5	20 µM	V2M K245T		V72I* L435R P445S			
6	VC matched with 20 µM	K245T	T176D C177I/F	L435R			
1	40 µM	K245T		V72I*			
2	40 µM	K245T E315G		T27S V72I* L435R		Q172Y V173L L175S R187Q	V360L V607I V832D * T833I K834C W835R
3	VC matched with 40 µM	K245T		V72I * L435R			E901D C902I
4	40 µM	K245T	F9L			Y87H	
5	40 µM	K245T	N98S	V72I * L435R		Y87H	V360L V607I V832D *
6	VC matched with 40 µM	K245T		V72I * L435R			
1	80 µM	K245T		V72I*		Y87H	V360L V607I
2	80 µM	K245T E315G		T27S, V72I * L435R			V360L V607I V832D * T833I K834C W835R
3	VC matched with 80 µM	K245T	T176D C177I				E901D C902I
4	80 µM	K245T	F180L			Y87H	H289Y
5	80 µM	K227C K245T		L435R		Y87H Q172R T188I	V360L V607I V832D *
6	VC matched with 80 µM	K245T		V72I * E371D L435R			

* mutations detected by NGS in wild type viral stock.

**Table 3 ijms-22-02670-t003:** Molecular Mechanics Generalized Born Surface Area (MM-GBSA) theoretical affinity of sofosbuvir and UTP to ZIKV RdRp WT and V360L/V607I mutant. * Standard Error of the Mean.

Complex	ZIKV RdRp WT ΔE_b_ (kcal/mol) ± SEM *	ZIKV RdRp V360L/V607I ΔE_b_ (kcal/mol) ± SEM *
A-UTP	−87.84 ± 1.99	−87.52 ± 2.15
A-SOFOSBUVIR	−82.41 ± 2.13	−79.64 ± 1.16
C-SOFOSBUVIR	−74.24 ± 1.17	−79.40 ± 1.52
G-SOFOSBUVIR	−84.52 ± 1.64	−66.58 ± 1.79
U-SOFOSBUVIR	−87.72 ± 2.15	−70.78 ± 2.34

**Table 4 ijms-22-02670-t004:** Primer used to sequence the whole NS5 region. Coordinates are indicated on H/PF/2013 ZIKV strain (Gen Bank Accession Number KJ776791).

PRIMER	SEQUENCE	SENSE	GENE	From	To
**P822**	TGTGCCCATACACCAGCACTATGAT	forward	NS5	8218	8242
**P823**	GGGTCTCCTCTAACCWCTAGTCC	reverse	3′UTR	10,659	10,681
**P824**	TACTGGAACTCCTCYACAGCCAC	forward	NS4B	7554	7576
**P825**	CAATGATCTTCATGTTGGGAGC	reverse	NS5	8481	8502
**P826**	GTCTGYACCAAAGAAGAGTTCATCAAC	forward	NS5	8847	8873
**P828**	CAGTGRTCCTCGTTCAAGAATCCAAG	reverse	NS5	9135	9160
**P853**	CTTGGATTCTTGAACGAGGAYCACTG	forward	NS5	9135	9160
**P865**	GTTCTCCTCAATCCACACTCTGTT	reverse	NS5	10,113	10,136
**P866**	AACCTAGTGGTGCAACTCATTCG	forward	NS5	9513	9535

## Data Availability

Data are available on request.

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
