# Peer review of "Sofosbuvir Selects for Drug-Resistant Amino Acid Variants in the Zika Virus RNA-Dependent RNA-Polymerase Complex In Vitro"

_ijms, 2021, doi:10.3390/ijms22052670_

Round 1
Reviewer 1 Report
In this work authors have investigated the ZIKV resistance profile against sofosbuvir through cell-based in vitro selection experiments. This work may be considered after minor revision
Comments :
“In 2018 and 2019, no local mosquito-borne Zika virus transmission has been reported in the continental United States.” How about in 2020? Seem intro missing some recent information …
Line no 62: references 19-20 are old one, Could you refer some more recent and revise the intro with recent ones, work is exciting, as mentioned before, latest information about ZV might give some clear view on the current status.
Figure 1 : whats A,b,c d, add in figure and caption and discuss the same in the results and same in figure 2 .
Have you checked time dependant H-bond plot that will and compare the structures with initial and final, that may give more insights on binding stability and effect of mutants on stability.
Author Response
Response to Reviewer 1 Comments
In this work authors have investigated the ZIKV resistance profile against sofosbuvir through cell-based in vitro selection experiments. This work may be considered after minor revision.
Point 1: “In 2018 and 2019, no local mosquito-borne Zika virus transmission has been reported in the continental United States.” How about in 2020? Seem intro missing some recent information..
Response to Point 1: Unfortunately, we were not able to find the phrase “In 2018 and 2019, no local mosquito-borne Zika virus transmission has been reported in the continental United States” along the text. So, we cannot rephrase the expression, however we have searched on CDC web site (https://www.cdc.gov/zika/reporting/2020-case-counts.html) and also in 2020 there have been no confirmed Zika virus disease cases reported from U.S. territories.
Point 2: Line no 62: references 19-20 are old one, Could you refer some more recent and revise the intro with recent ones, work is exciting, as mentioned before, latest information about ZV might give some clear view on the current status.
Response to Point 2. As suggested by the reviewer, we have updated the references concerning the ZIKV drugs in development (line 62) adding a recent review (PMID 33369711) and a paper on the development of compounds inhibiting a cellular target (PMID 32690969). Moreover, we updated the number of vaccines in development (line 56).
Point 3: Figure 1: whats A,b,c d, add in figure and caption and discuss the same in the results and same in figure 2.
Response to Point 3: We thank the reviewer for his/her comment, the content of panels A, B C, D in figure 1 and figure 2 has been added to the respective figure’s caption, (see lines 646-648, and 654-657). Likewise, the individual panels of figure 1 have been explicitly cited in the main text, Results section (see line 129).
Point 4: Have you checked time dependant H-bond plot that will and compare the structures with initial and final, that may give more insights on binding stability and effect of mutants on stability.
Response to Point 4: We agree with the reviewer that a time-dependent analysis of ligands interactions in MD simulations might give more insights into the effect of the mutant. We carried-out different types of time-dependent analyses to monitor: i) the total number of H-bonds established by the ligands (UTP and sofosbuvir); ii) the number of H-bonds established in the base-ligand pairing of each system; iii) ligands stability along the MD trajectories (RMSD). Although there is a general consensus showing a stronger stabilization of the wild type compared to the V360L/V607I mutant ZIKV RdRp, particularly in some systems, these results are not so clear-cut compared to those observed in the calculation of the theoretical affinity. In our opinion, and based on the binding poses provided by MD simulations, this might be explained by considering that ligand binding to the RdRp catalytic site is a complex process involving multiple players, such as Mg2+ ions, the nascent RNA strand, the template RNA strand, and amino acids from the catalytic site. Also, different types of intermolecular interactions are involved, such as electrostatic, metal coordination, hydrophobic, aromatic, and H-bond interactions. Thus, even if H-bonds are important in the base-ligand pairing, multiple contributions participate in the ligand’s theoretical affinity. The time-dependent analysis of individual contributions does not provide a clear and complete picture of the ligand-binding interactions and might lead the reader to a wrong channel of information. For these reasons, we decided not to include them in the manuscript.
We wish however to thank the reviewer for this stimulating comment.
Reviewer 2 Report
The authors show that V360L and V607I mutations of Zika virus NS5 decrease sofosbuvir susceptivity for zika virus and confirmed that by molecular docking and MD simulations. This study is important and will help us to develop new broad-spectrum antiviral drugs, especially for flavivirus. The authors have previously published a paper about sofosbuvir low-sensitive WNV isolation. The previous paper and this manuscript came to the same conclusion using roughly the same methodology except for the different viruses. Thus, I think that further analysis is needed to publish this manuscript.
1. The authors identified V360L and V607I mutations that decrease the susceptivity of sofosbuvir by serial virus passage. This "forward" approach is a standard method to identify mutations that affect drug susceptivity. To confirm the role of identified mutations, the "reverse" approach is also a standard method. I think that analyzing the role of V360L and/or V607I by "reverse" approach is necessary to conclude that only V360L and V607I decrease the susceptivity of sofosbuvir. Making recombinant virus containing the mutations is the best but measuring the activity of a polymerase containing the mutations will also give the minimum necessary results.
2. The authors showed IC50 of low-susceptive viruses for sofosbuvir but did not show titer of the viruses. The growth kinetics of mutant virus with and without the drug is one indicator when considering drug resistance. The data that growth kinetics of the virus containing V360L and V607I is useful for the discussion of the trade-off between viral propagation and drug resistance.
3. The authors hypothesized that the drug might be included in the nascent RNA regardless of the nucleotide in the RNA template and showed that by calculating the theoretical affinity of sofosbuvir and NTPs in wt and mutant RdRp. Can the mutation rate of wt and mutant RdRp be calculated from NGS data? If can, does the mutation rate support the theoretical affinity data?
Author Response
The authors show that V360L and V607I mutations of Zika virus NS5 decrease sofosbuvir susceptibility for zika virus and confirmed that by molecular docking and MD simulations. This study is important and will help us to develop new broad-spectrum antiviral drugs, especially for flavivirus. The authors have previously published a paper about sofosbuvir low-sensitive WNV isolation. The previous paper and this manuscript came to the same conclusion using roughly the same methodology except for the different viruses. Thus, I think that further analysis is needed to publish this manuscript.
Point 1: The authors identified V360L and V607I mutations that decrease the susceptibility of sofosbuvir by serial virus passage. This "forward" approach is a standard method to identify mutations that affect drug susceptibility. To confirm the role of identified mutations, the "reverse" approach is also a standard method. I think that analyzing the role of V360L and/or V607I by "reverse" approach is necessary to conclude that only V360L and V607I decrease the susceptibility of sofosbuvir. Making recombinant virus containing the mutations is the best but measuring the activity of a polymerase containing the mutations will also give the minimum necessary results.
Response to Point 1: We basically agree with the reviewer’s comment. Indeed, we planned to perform these analyses. Unfortunately, because of the SARS-COV-2 pandemic we had serious problems to access the BSL-3 facilities due to the priority to manage the pandemic. The biochemistry group also had problems to access the laboratory to generate and purify the polymerase. The current scenario makes it not possible to forecast when routine access to the BSL3 facilities will be restored. Consequently, at this time we cannot do nothing but planning to perform these experiments in the future. Please note that in previous works describing the in vitro resistance selection of different compounds against flavivirus (i.e. PMID 31931104 and PMID 26694200), the emerging mutations were not analysed by generation of recombinant clones or a biochemical assay.
Point 2: The authors showed IC50 of low-susceptible viruses for sofosbuvir but did not show titer of the viruses. The growth kinetics of mutant virus with and without the drug is one indicator when considering drug resistance. The data that growth kinetics of the virus containing V360L and V607I is useful for the discussion of the trade-off between viral propagation and drug resistance.
Response to Point 2: All the viruses were harvested at 80% CPE, which makes their titres comparable. The impact of NS5 mutations on the replicative capacity can be appreciated by the time required for virus breakthrough. This information is given in table 1. In our opinion, adding the TCID50 data for all the harvested viruses would complicate table 1 while contributing information of limited value. However, the following lines indicate the experimental condition with SOF concentration (exp), the days post infection (DPI) when 80% CPE was observed, the NS5 mutations detected, the titer of the virus harvested:
Exp: 1 (40uM SOF); DPI at 80% CPE: 54; NS5: none; Titer: 4.64E+04 TCID50/ml
Exp: 1 (80uM SOF); DPI at 80% CPE: 18; NS5: V360L V607I; Titer: 1.69E+04 TCID50/ml
Exp: 2 (40uM SOF); DPI at 80% CPE: 34; NS5: V360L V607I; Titer: 5.69E+03 TCID50/ml
Exp: 2 (80uM SOF); DPI at 80% CPE: 22; NS5: V360L V607I; Titer: 1.61E+04 TCID50/ml
Exp: 3 (virus control for the 40uM SOF experiment); DPI at 80% CPE: 3; NS5: none; Titer: 1.81E+05 TCID50/ml
Exp: 3 (virus control for the 80uM SOF experiment); DPI at 80% CPE: 3; NS5: none; Titer: 2.54E+03 TCID50/ml
Exp: 4 (40uM SOF); DPI at 80% CPE: 34; NS5: H289Y; Titer: 1.54E+04 TCID50/ml
Exp: 4 (80uM SOF); DPI at 80% CPE: 19; NS5: H289Y; Titer: 6.63E+04 TCID50/ml
Exp: 5 (40uM SOF); DPI at 80% CPE: 41; NS5: V360L V607I; Titer: 1.09E+04 TCID50/ml
Exp: 5 (80uM SOF); DPI at 80% CPE: 23; NS5: C269Y V360L V607I; Titer: 1.61E+04 TCID50/ml
Exp: 6 (virus control for the 40uM SOF experiment); DPI at 80% CPE: 3; NS5: none; Titer: 8.09E+03 TCID50/ml
Exp: 6 (virus control for the 80uM SOF experiment); DPI at 80% CPE: 3; NS5: none; Titer: 8.09E+03 TCID50/ml
Please note that we have modified tables 1 and 2 to clarify that there was no sofosbuvir pressure in the “virus control” experiments.
Point 3: The authors hypothesized that the drug might be included in the nascent RNA regardless of the nucleotide in the RNA template and showed that by calculating the theoretical affinity of sofosbuvir and NTPs in wt and mutant RdRp. Can the mutation rate of wt and mutant RdRp be calculated from NGS data? If can, does the mutation rate support the theoretical affinity data?
Response to Point 3: We thank the reviewer for this intriguing comment. We derived the mutation rates from NGS data both in the presence and absence of sofosbuvir. The mutation rate apparently increases with the sofosbuvir concentration but normalization of data per number of days of culture clearly shows that time of culturing drives the rate. The rate of mutations per day ranges from 7.7E-06 to 1.8E-05 for cultures in the presence of sofosbuvir and from 1.1E-05 to 4.6E-05 for control cultures in the absence of sofosbuvir. This is in line with lack of evidence that sofosbuvir exerts any mutagenic effect. In addition, we analysed the possible bias for specific mutations in the presence vs. absence of sofosbuvir by calculating the ratio between the relative frequency of the individual mutations in the two experimental conditions. On average, the mutation rate was 2.45-fold higher with sofosbuvir, again reflecting longer culturing. However, there were differences in the ranking of mutations by preference, ranging from T->G scoring at 0.58-fold to T->C scoring at 4.75-fold:
Mutation: T>C; fold: 4.75
Mutation: G>A; fold: 4.00
Mutation: C>T; fold: 3.80
Mutation: G>T; fold: 3.14
Mutation: A>C; fold: 2.73
Mutation: A>G; fold: 2.63
Mutation: A>T; fold: 1.67
Mutation: C>A; fold: 1.43
Mutation: T>A; fold: 1.33
Mutation: T>G; fold: 0.58
While the width of the range of the calculated fold values may be interesting, at this stage we prefer not to include this part in the revised paper feeling it would be too theoretical and not add substantial evidence.
Round 2
Reviewer 2 Report
The authors have responded to all my comments.